# Multiple-Reaction Monitoring Tandem Mass Method for Determination of Phenolics and Water-Soluble Vitamins in *Eccoilopus formosanus*

**DOI:** 10.3390/molecules25163632

**Published:** 2020-08-10

**Authors:** Ho-Shin Huang, Hsu-Sheng Yu, Chia-Hung Yen, Ean-Tun Liaw

**Affiliations:** 1Food Safety Center Laboratory, Golden Crops Corporation, Yun Lin 640, Taiwan; adinol.huang@gmail.com; 2Department of Food Science, National Pingtung University of Science & Technology, Pingtung 91201, Taiwan; hsyu@mail.npust.edu.tw; 3Department of Biological Science and Technology, National Pingtung University of Science and Technology, Pingtung 91201, Taiwan; chyen0326@mail.npust.edu.tw

**Keywords:** *Eccoilopus formosanus*, multiple-reaction monitoring, phenolics

## Abstract

This study established a validated method for the quantitative and qualitative determination of eight signature compounds in *Eccoilopus formosanus*. We used multiple-reaction monitoring scanning for quantification, and switched the electrospray ion source polarity between positive and negative modes in a single chromatographic run. The precursor-to-product ion transitions were *m/z* 355/163, *m/z* 181/163, *m/z* 265/122, *m/z* 269/117, *m/z* 170/152, *m/z* 377.2/180.7, *m/z* 169/124.8 and *m/z* 193/134 for chlorogenic acid, caffeic acid, thiamine, apigenin, pyridoxamin, riboflavin, gallic acid and ferulic acid, respectively. The developed method was also validated for accuracy, precision and limit of quantification. In this method, eight compounds were quantified with correlation coefficients of greater than 0.995. A high recovery (81.5–94.1%) and good reproducibility was obtained for five phenolics and three vitamins with the relative standard deviation, ranging from 1.2 to 3.5%. This method may be applied to the determination of both phenolics and water-soluble vitamins in cereal grain. The results may suggest that the extract of *E. formosanus* could be a good source of bioactive phytochemicals.

## 1. Introduction

Recently, global climate change, along with extreme weather conditions, has become a major concern for modern civilization. The situation is aggravated by continuous population growth, which, coupled with the other issues, is leading to the imminent food crisis. Fortunately, communities situated near biodiverse and rich areas have an abundant supply of potential foods, such as millets. Millets are small seeded grains with different varieties, which belong to several plant taxonomic groups and serve as a major food component in many Asian populations. Millets are rich sources of several phytochemicals, such as vitamins and minerals, and offer several health benefits [1,2,3]. B-vitamins in food are essential for the normal development of body functions and a group of water-soluble vitamins that play important roles in cell metabolism. Phenolics, a group of secondary metabolites, exist as free or glucoside forms in plants. Owing to their strong antioxidative activities, phenolic compounds are often incorporated into health foods for protection against cardiovascular disease.

Taiwanese oil millet (*Eccoilopus formosanus*) is a small seeded grain (Figure 1) and, as it is synonymous with Poaceae, it is able to survive droughts, colds and salinized soil. This species is easier to grow than most major high yield crops, such as rice, wheat and corn, which are reliant on heavy irrigation, herbicides and fertilization. However, to the best of our knowledge, the extraction of phenolics and water-soluble vitamins from *Eccoilopus formosanus* and its α-glucosidase inhibitory activity have never been evaluated. The most commonly used methods for the measurement of vitamins and phenolics are high-performance liquid chromatography (HPLC) methods using UV, but their low sensitivities and uncertainties in peak identifications limit their usage in analyses. Recently, tandem mass spectrometric detection has led to an improvement in detectability and selectivity by employing multiple-reaction monitoring (MRM) to analyze bioactive compounds [4,5,6]. In this study, a validated and simple HPLC–MS/MS with an MRM mode method was developed for the simultaneous quantitation of various phenolics and B-vitamins in Taiwanese oil millet. In addition, we evaluated the in vitro antioxidant and α-glucosidase inhibitory activity of an extract of *E. formosanus.*

## 2. Results and Discussion

### 2.1. Analysis of Eight Signature Compounds by High-Performance Liquid Chromatography Electrospray Ionization Mass Spectrometry with Multiple Reaction Monitoring

High-performance liquid chromatography electrospray ionization mass spectrometry (HPLC-ESI-MS, Applied Biosystem, Foster City, CA, USA) with the multiple reaction monitoring method was carried out for the qualitative and quantitative determination of the phenolic and soluble vitamins compounds from *E. formosanus*. The RP-C18 column (XBridge BEH 2.5 μm, 150 mm × 3 mm, Waters, Ireland) was selected for the HPLC-ESI-MS analysis based on its good separation ability and short analytical time. With the ESI source of the mass spectrometer, chlorogenic acid, caffeic acid, thiamine, pyridoxamin and riboflavin showed better sensitivity in the positive ion mode, but apigenin, gallic acid and ferulic acid exhibited a higher sensitivity in the negative ion mode (Figure 2). Given this discrepancy, a positive–negative conversion multiple reaction monitor was used to determine the content of the above eight compounds. The precursor-to-product ion transitions were *m/z* 355/163, *m/z* 181/163, *m/z* 265/122, *m/z* 269/117, *m/z* 170/152, *m/z* 377.2/180.7, *m/z* 169/124.8 and *m/z* 193/134 for chlorogenic acid, caffeic acid, thiamine, apigenin, pyridoxamin, riboflavin, gallic acid and ferulic acid, respectively. Their optimized declustering potential (DP), collision energies (CE) and collision cell exit potentials (CXP) are listed in Table 1.

Eight compounds showed excellent linearity with R^2^ > 0.995 and the results are shown in Table 2. The limits of quantification (LOQs) of eight compounds were 1.6, 0.9, 1.2, 0.3, 0.2, 0.1, 1.5 and 0.5 μg/mL for chlorogenic acid, caffeic acid, thiamine, apigenin, pyridoxamin, riboflavin, gallic acid and ferulic acid, based on a signal-to-noise ratio of 10:1, respectively. A high recovery (81.5–94.1%) and precision was acceptable, and relative standard deviation (RSD) values ranged between 1.2% and 3.5% for intra-day variation. This validated method was successfully applied to the quality control of the *E. formosanus* extract, which provided particularly important information for production and application.

The previous methods of analyzing B-vitamins, including microbiological, radioactive, enzyme and high-performance liquid chromatography were used for analyses, but could be used only for individual and time-consuming pretreatments [7,8,9]. In this study, a simple, sensitive and easy extraction LC–MS/MS method was validated for the analysis of *E. formosanus*. The MRM chromatographic profiles of the extract from *E. formosanus* are shown in Figure 3. The described HPLC-ESI-MS method was applied to the analyses and quality evaluation of *E. formosanus* through a simultaneous determination of phenolics and water-soluble vitamin compounds. The quantitative analytical results (Table 3) indicated that their contents were distributed in these samples. The contents of chlorogenic acid, caffeic acid, thiamine, apigenin, pyridoxamin, riboflavin, gallic acid and ferulic acid in the extract were in the range of 1.5–179.1 μg/g. In these results, chlorogenic acid and ferulic acid had the highest (179.1 μg/g) and second highest (145.3 μg/g) abundance, respectively, among the eight compounds that were simultaneously obtained. The previous study indicated that cereal bran is a major source of phenolic acids and antioxidants [10], with ferulic acid as its major antioxidant [11]. This method allows for the simultaneous monitoring of retention time of the isolated compound peak and the determination of MS data during analysis. We developed an accurate quantification of eight signature compounds. The overall time required for chromatographic separation is within 30 min.

### 2.2. The Antioxidant Activity, α-Glucosidase Inhibition Activity and Content of Total Polyphenols

The total phenolic contents of the *E. formosanus* extract were expressed as gallic acid equivalents. The total phenolic contents of *E. formosanus* were determined using different concentrations of ethanol as a solvent and an ultrasonic bath for at least one hour of extraction. In total, 50% ethanol resulted in a high yield, and the total polyphenols were 45.5% and 3.1 mg gallic acid equivalents /g, respectively (Table 4). The radical scavenging activities of the *E. formosanus* extract and individual compounds were evaluated using DPPH (2, 2 diphenyl-1-picrylhydrazyl) radical scavenging assays (Table 5). The ethanol extract from *E. formosanus* exhibited stronger DPPH scavenging with an IC_50_ value of 120.7 ± 0.3 μg/mL, which had an effective capacity for scavenging for superoxide radicals and was correlated with phenolics and water-soluble vitamins. The concentration effect curves are presented in Appendix A.

Some previous reports have proven that the consumption of grain could reduce the risk of cardiovascular disease and could be linked to bioactive compounds, such as polyphenols and vitamins [12]. In this study, the *E. formosanus* extract showed the presence of chlorogenic acid, caffeic acid, thiamine, apigenin, pyridoxamin, riboflavin, gallic acid and ferulic acid. Some phenolics have been shown to exhibit inhibitory effects against α-glucosidase enzymes [13]. Table 5 shows the α-glucosidase inhibition activity of individual compounds and its concentration effect curves are presented in Appendix A, Appendix A. The α-glucosidase inhibition activity of *E. formosanus* might be attributed to its bioactive compounds; however, more extensive studies, such as in vivo tests, should be performed to elucidate their mechanisms of action in the future.

## 3. Materials and Methods

### 3.1. Reagents

HPLC-Grade ethanol, acetonitrile and chemicals were purchased from Merck, Mumbai, India. Chlorogenic acid, caffeic acid, thiamine, riboflavin, apigenin, pyridoxamin, gallic acid, ferulic acid, p-nitrophenyl-α-d-glucopyranoside and α-glucosidase were purchased from Sigma-Aldrich (St Louis, MO, USA).

### 3.2. Plant Collection, Identification and Extraction

The whole seed part of *Eccoilopus formosanus* was collected in March 2020 from the Kaohsiung District Agricultural Research and Extension Station, Taiwan. The plant material was botanically identified by Supervisor Kuo-lung, Chou, from the Kaohsiung District Agricultural Research and Extension Station. The voucher specimen was deposited in the Laboratory of Functional Food of National Pingtung University of Science and Technology, and the plant code is 2020EF01. Later, the whole plant was washed and cleaned, then dried in the shade at room temperature, with continuous overturning to prevent mold. Next, the plant was weighed, ground using a mechanical grinder and the materials were placed in airtight bottles and stored in the desiccators to be used for later extraction. The plant materials were washed with water and air-dried at room temperature for 3 days. The collected sample was then oven-dried at 50 °C for 3 days and ground to a fine powder by an electronic blender. The dried *E. formosanum* (50 g) was mechanically ground to a fine powder and then sieved through a 10-mesh sieve. The obtained plant powder material was extracted with different concentrations of ethanol (30%, 50% and 70%) at room temperature for more than one hour by ultrasonication and was then concentrated until it became dry powder.

### 3.3. HPLC-ESI-MS Analysis of the Extract

Quantitative LC–MS/MS analysis was performed using a Nexera XR-20A system (Shimadzu 8045, Kyoto, Japan) coupled to an API 4000 triple quadrupole tandem mass spectrometer (Applied Biosystem, Foster City, CA, USA). Chromatographic separation was performed on an XBridge BEH C18 column (150 mm × 3.0 mm I.D, 2.5 μm; Waters, Ireland). The mobile phase consisted of 0.1% formic acid aqueous solution (solution A) and acetonitrile (solution B) and a gradient elution program was set as follows: solution A, 90–60% (0–7.5 min); 60–40% (7.5–10.8 min); 40–0% (10.8–19 min); 0–40% (19–21 min); 40–60% (21–25 min) and 50–90% (25–30 min). The column temperature was fixed at 40 °C, the flow rate was set at 0.3 mL/min and the injection volume was 1 μL. The electrospray negative mode was selected as an ion source for gallic acid, apigenin and ferulic acid detection. The positive electrospray mode was selected as an ion source for chlorogenic acid, caffeic acid, thiamine, pyridoxamin and riboflavin detection. The quantification was performed by multiple-reactions monitoring (MRM). The optimized ESI source parameters were as follows: ion spray voltage, −4500 V for negative mode and 4500 V for positive mode; nitrogen nebulizer gas pressure, 50 psi; nitrogen curtain gas pressure, 12 psi; heater temperature, 450 °C; collisionally activated dissociation (CAD) gas, 10 psi. The precursor-to-product ion transitions were *m/z* 355/163, *m/z* 181/163, *m/z* 265/122, *m/z* 377.2/180.7, *m/z* 269/117, *m/z* 170/152, *m/z* 169/124.8 and *m/z* 193/134 for chlorogenic acid, caffeic acid, thiamine, riboflavin, apigenin, pyridoxamin, gallic acid and ferulic acid, respectively. Their optimized declustering potentials (DP), collision energies (CE) and collision cell exit potentials (CXP) are listed in Table 1. All data acquisition and processing were performed using Analyst 1.6.3 software (AB SCIEX, Concord, ON, Canada).

### 3.4. Method Validation and Quantitation of Phenolic Acid and Water-Soluble Vitamins

The peak area of each component in the extract of *E. formosanus* was acquired from its chromatogram and the abundance of each compound was calculated from its corresponding calibration curve. Experiments were conducted in triplicate and the resulting data are expressed as the mean ± SE and the units are mg/g. The method was validated for linearity, accuracy, precision and limits of quantification (LOQs). Standards at the concentration range of 1–200 (μg/mL) were prepared. Solutions containing eight standards at five different concentrations were injected in triplicate. Linear regression equations were constructed by establishing a calibration graph with the peak area (y) and concentration (x, μg/mL). The mixed standards solution was further diluted to a certain concentration to explore the limits of quantification. The LOQs were determined at a signal-to-noise ratio of 10. The intra- and inter-day precisions were determined by continuously injecting the sample solution for three replicates on the same day and by measuring them once a day for three consecutive days. The recovery test for reflecting accuracy was conducted by the standard addition approach. The recovery yield was carried out according to the following formula: recovery yield (%) = [(amount detected−original amount)/amount spiked] × 100%, and RSD (%) = (SD/mean) × 100%. The repeatability was estimated on the grounds of relative standard deviation (RSD).

### 3.5. Determination of Total Polyphenol Content

The total phenolic content (TPC) of the extract was determined using the Folin–Ciocalteu reagent [14]. In brief, a solution of extract (0.5 mL) with proper dilution was mixed with 1.0 mL of the Folin–Ciocalteu reagent at room temperature (5 min) and added to 2 mL of the 7% Na_2_CO_3_ solution. The mixture was boiled for 1 min and the absorbance of the color was recorded at 750 nm in a spectrophotometer (Shimadzu, Kyoto, Japan). The results are expressed in gallic acid equivalent (GAE)/g, on a dry weight basis.

### 3.6. DPPH Radical Scavenging Activity

The scavenging activity of the extract against the DPPH (2, 2 diphenyl-1-picrylhydrazyl) radical was assessed according to a reference method with some modifications [15]. In brief, 0.1 mL of extract was mixed with 3.0 mL of DPPH solution (0.1 mM). The reaction mixture was left in the dark at room temperature for 30 min. The absorbance of the mixture was measured at 517 nm. The percentage of scavenging activity against the DPPH radical was calculated by the following equation:Radical scavenging activity (%) = (Abs control − Abs sample)/Abs control × 100%
where Abs control is the absorbance of the DPPH radical in methanol; Abs sample is the absorbance of the DPPH radical solution mixed with the sample extract. The IC_50_ value, indicating the concentration at which a sample would inhibit free radicals by 50%, was also calculated. All determinations were performed in triplicate.

### 3.7. α-Glucosidase Inhibition

The assay of α-glucosidase inhibition was performed according to a reference method with some modifications [16]. In brief, 100 µL phosphate buffer (50 mM, pH 6.5) and a 50 µL extract were mixed with 50 µL α-glucosidase (1.5 U/mL in phosphate buffer). The enzyme reaction was initiated by adding 100 µL p-nitrophenyl-α-d-glucopyranoside (1 mM in phosphate buffer) and the mixture was incubated for 30 min at 37 °C. The enzyme reaction activity was terminated by the addition of 100 µL Na_2_CO_3_ (0.1 M). Enzyme activity was measured at 405 nm. The mixture of all other reagents and the enzyme, except the test sample, was used as a blank, while the mixture without the test sample and enzyme was taken as a control. The percentage of scavenging activity against DPPH radical was calculated by the following equation:Percentage of enzyme inhibition (%) = (Abs control − Abs sample)/Abs control × 100%
where Abs control is the absorbance in ethanol; Abs sample is the absorbance with sample extract. The IC_50_ value was calculated to determine the 50% inhibitory capacity of the reaction at a certain concentration. All determinations were performed in triplicate.

## 4. Conclusions

We successfully developed a precise and accurate HPLC-ESI-MS method to determine the eight major signature compounds in *E. formosanus*. Based on the validation results, the developed method can be useful for the detection of eight signature compounds (chlorogenic acid, caffeic acid, thiamine, riboflavin, apigenin, pyridoxamin, gallic acid and ferulic acid) under the specified conditions. This method may be applied to the determination of both phenolics and water-soluble vitamins in cereal grain.

## Figures and Tables

**Figure 1 molecules-25-03632-f001:**
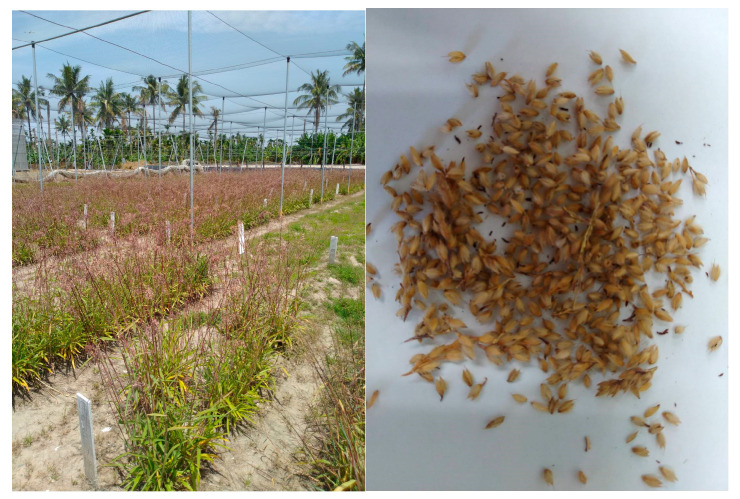
The appearance of *Eccoilopus formosanus*.

**Figure 2 molecules-25-03632-f002:**
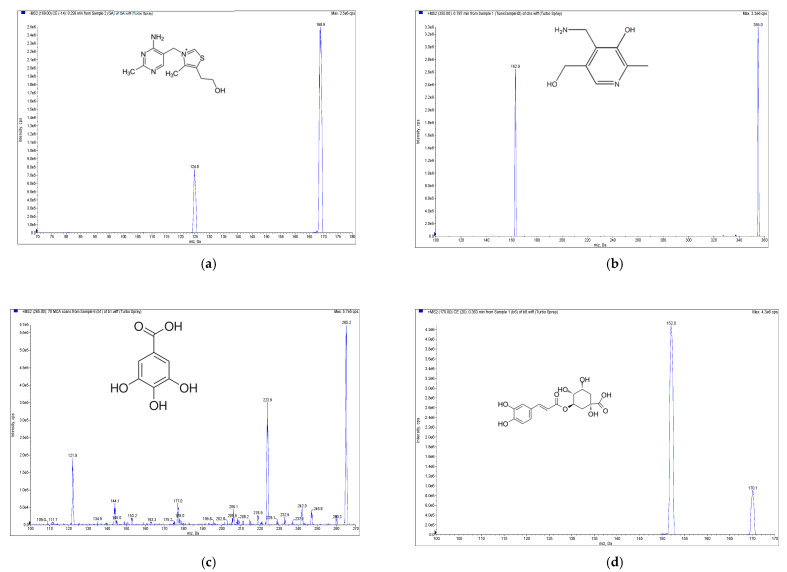
Mass spectrometry (MS) spectra of the extract of *E. formosanus*. (**a**) Thiamine, (**b**) pyridoxamin, (**c**) gallic acid, (**d**) chlorogenic acid, (**e**) caffeic acid, (**f**) riboflavin, (**g**) ferulic acid and (**h**) apigenin.

**Figure 3 molecules-25-03632-f003:**
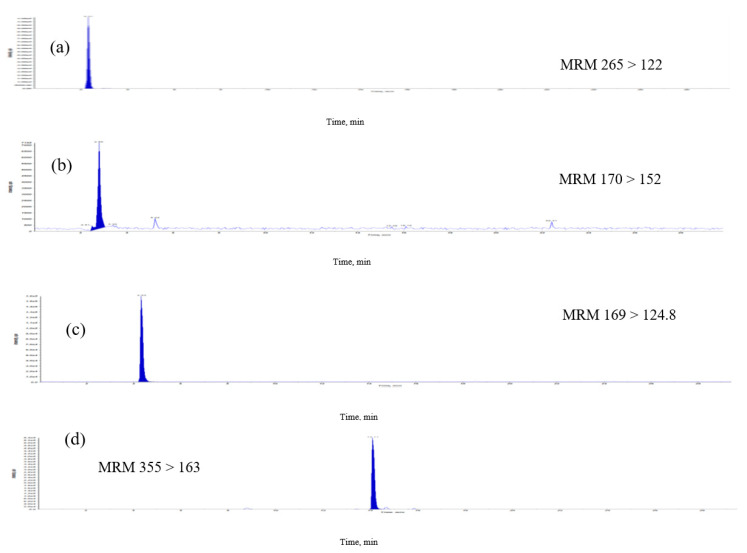
Multiple-reaction monitoring (MRM) chromatograms of extract of *E. formosanus*. (**a**) Thiamine, (**b**) pyridoxamin, (**c**) gallic acid, (**d**) chlorogenic acid, (**e**) caffeic acid, (**f**) riboflavin, (**g**) ferulic acid and (**h**) apigenin.

**Table 1 molecules-25-03632-t001:** Multiple-reaction monitoring (MRM) transitions, declustering potential (DP), collision energy (CE) and collision cell exit potential (CXP) of eight signature compounds.

Compound	MRM Transition (*m/z*)	DP (V)	CE (eV)	CXP (V)
Thiamine	265 >122	62	22	18
Pyridoxamin	170 > 152	70	28	7
Gallic acid	169 > 124.8	−29	−23	−27
Chlorogenic acid	355 >163	66	14	10
Caffeic acid	181 >163	45	17	9
Riboflavin	377.2 >180.7	50	20	10
Ferulic acid	193 > 134	−36	−20	−4.5
Apigenin	269 > 117	−90	−52	−15

**Table 2 molecules-25-03632-t002:** Linear regression equations, coefficients (R^2^), linear range, precisions, recovery yields and limit of quantification (LOQ) of eight signature compounds.

Compound	Retention Time (tR)	Regression Equation	R^2^	Range (μg/mL)	LOQ (μg/mL)	Precision (%)	Recovery (%)
Thiamine	2.32	y = 9062.1x−2446.1	0.995	1–50	0.3	2.3	94.1
Pyridoxamin	2.81	y = 2191.6x−695.89	0.995	1–50	0.2	1.7	81.5
Gallic acid	4.30	y= 2915.1x−3240	0.996	1–50	1.2	3.5	90.5
Chlorogenic acid	14.08	y = 1058.3x−60.5	0.997	10–200	1.6	1.5	91.2
Caffeic acid	15.39	y = 5695.4−1165.5	0.999	1–50	0.9	1.4	86.7
Riboflavin	15.66	y =1415.8+1248.6	0.995	1–50	0.1	1.2	85.5
Ferulic acid	18.34	y = 934.5−2352.1	0.996	10–200	1.5	1.9	82.3
Apigenin	23.98	y = 3276.6x−658.1	0.997	1–50	0.5	2.9	92.1

**Table 3 molecules-25-03632-t003:** The contents (μg/g) of the eight signature compounds in the extract from *E. formosanus* (*n* = 3).

Sample	Thiamine	Pyridoxamin	Gallic Acid	Chlorogenic Acid	Caffeic Acid	Riboflavin	Ferulic Acid	Apigenin
Extract of *E. formosanus*	4.6	1.5	11.0	179.1	65.2	2.1	145.3	15.2

**Table 4 molecules-25-03632-t004:** Extract yield and total phenolic content of *E. formosanus* with different ethanol concentration extractions.

Samples	Yield (%)	Total Phenolic Content (mg GAE/g, DW)
30% ethanol	37.1 ± 0.3	2.5 ± 0.3
50% ethanol	45.5 ± 0.2	3.1 ± 0.1
70% ethanol	40.4 ± 0.6	2.9 ± 0.5

Data values represent the mean ± SD (*n* = 3), GAE-gallic acid equivalents, DW-dry weight.

**Table 5 molecules-25-03632-t005:** Antioxidant activities and α-glucosidase inhibition activity of *E. formosanus* extract.

Samples	DPPH Radical Scavenging Activity, IC_50_ Value (µg/mL)	α-Glucosidase Inhibition Activity, IC_50_ Value (µg/mL)
*E. formosanus* extract	120.7 ± 0.3	136.9 ± 0.5
Thiamine	>500	>500
Pyridoxamin	>500	>500
Gallic acid	21.1 ± 0.3	59.2± 0.6
Chlorogenic acid	12.7 ± 0.5	9.1 ± 0.5
Caffeic acid	25.3 ± 0.3	7.2 ± 0.2
Riboflavin	155.7 ± 0.4	>500
Ferulic acid	45.2 ± 0.3	107.1 ± 0.9
Apigenin	31.6 ± 0.9	79.1 ± 0.6

Data values represent the mean ± SD (*n* = 3)

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
