# Peer review of "Multiple-Reaction Monitoring Tandem Mass Method for Determination of Phenolics and Water-Soluble Vitamins in Eccoilopus formosanus"

_molecules, 2020, doi:10.3390/molecules25163632_

Round 1
Reviewer 1 Report
The authors reported an article: Multiple Reaction Monitoring Tandem Mass Method for Determination of Phenolics and Water-Soluble Vitamins in Eccoilopus formosanus. The data presented in this paper is important to understand that by changing the ionization energies it is possible to adjust the mass spectrometry in order to successfully quantify the chemical composition of the extracts extracted from Eccoilopus formosanus with ethanol.
The strengths are certainly the use of chromatographic method together with Tandem Masspectrometry in the analysis of natural substances.
The weaker points;
I did not see in the manuscript were about the extraction process and the best parameters. How the criteria were set to determine the best extraction conditions?
Obtained extract concentration is not described clearly. Lyophilization was used or another method?
It would be necessary to show more about sample preparation and storage.
Secondly, the extraction yields, how many extraction experiments were performed, and the criteria according to which the best parameters were determined.
The total phenolic content (TPC) of the extract is suspiciously low. There are no mistake?
Author Response
We'd like to appreciate your kind care on our manuscript and all the excellent comments and suggestion. We have modified the manuscript according to your revisions. The revisions are heightened in red color in our revised manuscript. We have evaluated the extraction yields and total phenolic content of different extraction parameters in this study (table 4). Once again, thank you so much for your effort to our research work.

Reviewer 2 Report
This Manuscript needs to be proof read by a native english speaker. The value of the science is greatly diminished by the incoherent language used. I am more than happy to review this manuscript again once it has been clearly written
Author Response
We'd like to appreciate your kind care on our manuscript and all the excellent comments and suggestion. This paper has undergone English language editing by experienced and native English speaking editors of MDPI. The revision are heightened in pink color in our revised manuscript. Once again, thank you so much for your effort to our research work.

Reviewer 3 Report
In this study a LC/MS-MS method was developed and validated for the simultaneous quantification of 8 phenolic compounds and water-soluble vitamins in Eccoilopus formosanus. The method was described adequately and the results indicated that the method was repeatable and sensitive. I just have several minor comments.
(1) Line 112-113 "We developed a rapid and accurate quantification of eight signature compounds." As shown in Table 2, the retention time for Apigenin is about 24 minutes. This is not considered a "rapid" method. Have the authors tried to use a shorter colume (e.g., 50 mm) to reduct the run time? Or was there any particular reason that separation of those compounds needed to be achieved?
(2) What is the X-axis of Figure 3? Is it the time scale? The authors should include a chromatogram with all 8 compounds on it.
(3) It was disappointing to see that the IC50 values presented in Table 4 were for the extracts. Since this study was focused on the LC/MS-MS method development and validation, the authors should demonstrate the significance of their method by determining the IC50 values for individual compounds. The log-concentration versus effect graphs should be included in the manuscript as well.
Author Response
We'd like to appreciate your kind care on our manuscript and all the cexellent comments and suggestion. We have modified the manuscript point by point according to your revisions.
- We have chosen to delete the word about " rapid " in the manuscript.
- We have added the " Time, min " on the X-axis of each graph of Figure 3.
- We have added the IC50 test of each compound and listed the results in Table 5. In addition, the concentration and inhibition curves was presented in supplementary figure S1 and S2. Once again, thank you so much for your effort to our research work.

Round 2
Reviewer 2 Report
Huang H-S et al have described a valid and clear MRM tandem mass method for the quantitative determination of known phenolics and vitamins in E. formosanus. The experiments conducted and results acquired are sound and would be of high interest to the readers in Food Science chemistry.